# OpenReview forum: "Explore, Establish, Exploit: Red Teaming Language Models from Scratch"
_ICLR.cc/2024/Conference — Submitted to ICLR 2024_

### Official Review · Reviewer_tVNe · 2023-10-24

**Soundness:** 3 good
**Presentation:** 4 excellent
**Contribution:** 2 fair
**Rating:** 5
**Confidence:** 4

**Summary:**

The authors propose a 3-step human-in-the-loop red teaming framework that can red team LMs from scratch (without any previously given red team classifiers).
The proposed framework consists of the following 3 steps:
1. Explore LM behaviors and collect data.
2. Human-label data from the first step and train a red team classifier using these data.
3. Train an RL agent to generate adversarial prompts, which elicit responses from the model that are classified as harmful.

The experimental results show that the proposed method can successfully red-team GPT-3 and GPT-2-xl models.
Moreover, the authors produce the CommonClaim dataset consisting of 20,000 completions by GPT-3-text-davinci-002 human labelled as 'true', 'false', or 'neither'.

**Strengths:**

- The paper is well-written and easy to understand
- The idea of the paper is simple and clear.
- The proposed method adapts diversity sampling techniques during exploration and exploitation steps to enhance the diversity of the resulting adversarial prompts. The empirical results show that diversity sampling techniques are key to avoiding mode collapses.
- The authors contribute to society by providing the CommonClaim dataset.

**Weaknesses:**

- Lack of novelty
- Lack of quantitative comparisons
- Limited applicability scenarios
- Some missing references

Please refer to the questions for the details.

**Questions:**

(Novelty) There exists a sort of study that tried to build the red team dataset and a classifier based on the human-in-the-loop framework [1,2,3]. These methods utilize human resources to generate adversarial prompts and label the harmfulness of the model response to construct a dataset and train a classifier. If I understood correctly, the main difference between existing studies can be written as following:
1. For the "Attack" part, the proposed "from scratch" framework utilized a rl-based attack algorithm instead of human attackers as in [1,2]. However, as stated in the paper, the rl-based attack algorithm has been used in prior works such as [Deng et al., 2022] or [Perez et al., 2022b].
2. The authors utilize diversity sampling to avoid mode collapses of the rl-based attack method. However, [Perez et al., 2022b] already emphasize the importance of diversity in red-teaming. Moreover, the other study proposed a red teaming approach that incorporates the diversity of adversarial prompts into the objective function throughout the red teaming process [4].

Can you clarify the novelty of the proposed "from scratch" red teaming method in detail?

(Quantitative Analysis) The paper provides a few quantitative analyses. Most of the experimental results are qualitative. The examples in Appendix B about mode collapse seems clear, but it would be more credible if you could provide quantitative red-teaming results with and without each diversity sampling in step 1 and 3 with appropriate diversity metric and performance metric.

(Limited Scenarios) I cannot agree with the justification of the scenarios. My questions can be divided into the following two parts:
1. In the paper, the authors state that "Most importantly, if failures can already be efficiently identified in advance, then red-teaming has limited value because bad text could simply be filtered from the model’s training data and/or outputs." However, filtering in training data or outputs can degrade the model performance. Model unlearning can be another solution to this problem, but it doesn't work well in my knowledge. Can you provide any references to your statement?
2. Once we made datasets and classifiers for some purposes like toxicity or fact-checking, we can re-use these again and again to red-team the different LMs. If you already have data and classifiers, it seems strange not to utilize them. Therefore, the situation, in which the proposed scenario is valid, is limited to cases where the target criteria of harmfulness are significantly different from the existing collected data, so the existing data cannot be utilized. I can't agree on whether there will be many cases like this, can you explain more? Or can you provide quantitative evidence that the proposed method is effective when there is a lot of data already collected and the classifier is learned?

(Missing References)
- [1] Build it Break it Fix it for Dialogue Safety: Robustness from Adversarial Human Attack, Dinan et al., 2019.
- [2] Bot-Adversarial Dialogue for Safe Conversational Agents, Xu et al., 2021.
- [3] SQuARe: A Large-Scale Dataset of Sensitive Questions and Acceptable Responses Created Through Human-Machine Collaboration, Lee et al., ACL 2023.
- [4] Query-Efficient Black-Box Red Teaming via Bayesian Optimization, Lee et al., ACL 2023.

**Details Of Ethics Concerns:**

The paper contains several subsections discussing the ethical perspectives associated with the research.

---

> ### Author Response · Authors · 2023-11-16
> **Response to tVNe**
>
> Thank you for the feedback and comments. We are glad to hear that you found the paper to be well-written and that the CommonClaim dataset will be valuable. Below are replies to weaknesses and questions.
>
> ### W1 & W4 – Re: Comparisons to prior work and missing references
>
> **New discussions of related work:** Thank you for the suggestions about prior work. In particular, we think that it will be useful for us to cite [Xu et al.  (2021)](https://aclanthology.org/2021.naacl-main.235/)([2]) as an example of the effectiveness of a filtering baseline. We added this. Meanwhile, we have added [Lee et al. (2023a)](https://arxiv.org/abs/2305.17696) [(3]) and [Ziegler et al. (2022)](https://arxiv.org/abs/2205.01663) as examples of human-generated attacks against language generators, and [Lee et al. (2023b)](https://arxiv.org/abs/2305.17444)([4]) next to our discussion of [Shi et al., (2022)](https://arxiv.org/abs/2212.10539) and [Kumar et al. (2022)](https://arxiv.org/abs/2205.12558) which use related langevin-dynamics-based approaches to automated red teaming.
>
> **In comparison to prior works that rely entirely on humans to generate adversarial prompts, our work is novel in its use of automated attacks in the Exploit step.** We agree that human-in-the-loop attacks are interesting and important. However, we believe that automated attacks are an important complement to these methods. In practice, we expect that a combination of both will be used. One advantage of our approach is that a fixed amount of human effort in the Establish step can be used to enable very large amounts of cheap, automated attacks. Another advantage is that sometimes it might be easier to define a target category of text than to attack a model to produce that target category. Also automated methods and humans simply tend to produce different types of prompts, so these different methods for attacking networks can be complementary.
>
> **Our attack method adds to the method used by [Perez et al. (2022)](https://arxiv.org/abs/2202.03286).** We are sorry that the paper was unclear about our contribution. Our diversity objective for RL is novel. While Perez et al. talk about the importance of diversity in RL-based attacks, they do not use our diversity objective (see, e.g., Figure 2 of their paper or the last paragraph of their section 2.2.) We have updated our discussion of the ablation experiments for this diversity term to clarify that the ablation experiment was with the same method as [Perez et al. (2022)](https://arxiv.org/abs/2202.03286) and [Deng et al. (2022)](https://arxiv.org/abs/2205.12548). We also added vocabulary size analysis to better quantify mode collapse. Across 100 samples, the prompt vocabulary size decreased from 116 to 3 and 81 to 9 for the toxicity and untruthfulness red teaming experiments without our diversity method.
>
> **We added a discussion of other prior techniques to improve diversity and noted that they are not applicable to RL attacks.** We added mention of how [Lee et al. (2023b)](https://arxiv.org/abs/2305.17444)([4]), [Shi et al., (2022)](https://arxiv.org/abs/2212.10539), and [Kumar et al. (2022)](https://arxiv.org/abs/2205.12558) use an analogous but non-RL technique for this. Thank you for pointing this out!
>
> **See also our global response.** We give an overview the paper’s novel contributions.
>
> ### W2 – Re: Quantitative analysis
>
> **We would like to clarify that we have a fair amount of quantitative evaluations of our method.** These include analysis of how much our attacks increased toxicity, mode collapse without our diversity technique, label agreement between humans and models, and the success of the prompt generators at eliciting harmful responses according to each classifier we used.  We discuss these in detail in the global response. Are there any other quantitative experiments you would like to see us add?
>
> **We are running a new experiment** in which we omit the diversity term used in the first step (Explore) for toxicity experiments. Thank you for the suggestion.

---

> > ### Author Response · Authors · 2023-11-16
> > **Response to tVNE Part II**
> >
> > ### W3.1 – Re: “However, filtering in training data or outputs can degrade the model performance.”
> >
> > **We might not fully understand this point.** Could you clarify in what way you meant this? If this means that someone might accidentally filter responses that they actually want if the classifier used to filter is a bad one, then we agree. This is one of the motivations of our work. We show that using a “bad” classifier off the shelf such as the CREAK classifier does a poor job at contextually classifying undesirable outputs from the target model and can end up being a poor filter. Were you referring to the additional compute costs of filtering?
> >
> > **Filtering train data and/or model outputs is common.** For example,  [Xu et al.  (2021)](https://aclanthology.org/2021.naacl-main.235/)([2]) which you brought to our attention advocates for doing both. Meanwhile, [Korbak et al. (2023](https://arxiv.org/abs/2302.08582) filter train data to make more aligned models, and [Phute et al., (2023)](https://arxiv.org/abs/2308.07308) show how filtering conversations at runtime can be a very effective way of defending against jailbreaks. We have added all of these to our discussion of this. Thank you for pointing these out.
> >
> > ### W3.2 – Re: “Once we made datasets and classifiers for some purposes like toxicity or fact-checking, we can re-use these again and again to red-team the different LMs”
> >
> > **We agree that if you already have good data and classifiers (emphasis on good), it seems strange not to utilize them. But in practice, this is often not the case.** Showing this was the goal of our experiment with CREAK, where we tried to red-team GPT3 with an off-the-shelf truthfulness classifier. This failed because it was not a good fit for the distribution of outputs that the model could produce. (In contrast, using the CommonClaim target we were able to elicit political statements.)
> >
> > **Our work connects to a growing body of work that emphasizes the importance of context in evaluation --- particularly in safety applications. We added to the paper to discuss this.** Two of the papers that you brought to our attention emphasize the value of contextuality. [Xu et al.  (2021)](https://aclanthology.org/2021.naacl-main.235/)([2]) mentions this, writing that “ the very definition of “safe" is both contextually and culturally dependent”. They also cite [Schmidt and Wiegand (2017))](https://aclanthology.org/W17-1101/). The poor transferability of classifiers because of how they struggle to adapt to context is also a central point in [Dinan et al. (2019)](https://arxiv.org/abs/1908.06083)([1]). They write “In this work we recommend that future systems should move beyond classification of singular utterances and use contextual information to help identify offensive language.” We added mention of these three works plus [Hendrycks et al., (2020)](https://arxiv.org/abs/2008.02275) to discuss the importance of contextuality when red teaming.
> >
> > Thank you again for your time and help! We are looking forward to the rest of the discussion period.

---

> ### Comment · Reviewer_tVNe · 2023-11-21
>
> Thank you for your kind responses. Global response and your responses sufficiently addressed most of my concerns on your paper.
>
> There was a line of research about detoxifying pre-trained language models [1,2,3].
> My previous comment “However, filtering in training data or outputs can degrade the model performance.” in W3.1 is based on [1], which shows that filtering toxic contents in training data can help detoxify the LM but also can result in performance degradation.
>
> I appreciate the inclusion of additional references, such as [4], in the revised version of the paper, which adequately addressed my concerns regarding this issue.
>
> In W3.2, I requested you to justify the "why from scratch" part.
> Your explanation, that the context of toxicity can be very diverse, was helpful in understanding the necessity.
> However, as you agreed, we still do not need to red-team a new model from scratch if we already have a dataset and a classifier sharing the same context.
> In this regard, I still think that the proposed red-teaming method has limited-applicability.
> For example, since the authors released a valuable dataset CommonClaim in this paper, we can red-team other new models for the purpose of fact-checking by re-using this dataset and classifier, not from scratch.
>
> Therefore, I still think that the proposed method will only be used in limited situations from the perspective of red teaming. However, from the perspective of utilizing the automatic red-teaming method to **collect datasets** and **train classifiers** corresponding to toxicity in certain new contexts, it seems like a nice research approach.
> In this regard, I raised my scores to 5.
>
> But, I think it would be better to emphasize the **collect datasets** and **train classifiers** parts, more than the **red-team** part in the paper.
>
>
> [1] Challenges in Detoxifying Language Models, Welbl et al., EMNLP findings 2021.
>
> [2] Detoxifying Language Models Risks Marginalizing Minority Voices, Xu et al., NAACL 2021.
>
> [3] Self-Diagnosis and Self-Debiasing: A Proposal for Reducing Corpus-Based Bias in NLP, Schick et al., TACL 2021.
>
> [4] Pretraining language models with human preferences, Korbak et al., ICML 2023.

---

> > ### Author Response · Authors · 2023-11-21
> > **Thanks + a note**
> >
> > Thank you for the reply. We appreciate your time. We are glad that we were able to address some concerns. Here is one note we would like to add before the discussion period ends.
> >
> > > However, as you agreed, we still do not need to red-team a new model from scratch if we already have a dataset and a classifier sharing the same context. In this regard, I still think that the proposed red-teaming method has limited-applicability.
> >
> > The point above is more a criticism of the entire literature on automated red-teaming methods than one of our work. It applies much less to this paper because we point out and confront this problem unlike prior works on automated attacks. As you point out, [Xu et al. (2021)](https://aclanthology.org/2021.naacl-main.235/) and other past work on manual red teaming has accounted for this problem, but we know of no prior papers that try to reconcile automated red teaming with this problem.
> >
> > If you are concerned with this issue in the literature on automated attacks, one reason to accept this paper would be that it is rather unique among others in the automated red teaming literature in making the point that if you already have the classifier and data, the problem is mostly solved already.
> >
> > Best,
> >
> > -authors

---

> > > ### Comment · Reviewer_tVNe · 2023-11-21
> > > **Thanks + Additional Comments**
> > >
> > > Thank you for getting back to me. Your last response helps me understand more about your work. Thank you!
> > >
> > > If I understood correctly, your work is in the line "human-in-the-loop toxic data construction method" [1,2,4]. According to your response, your work is distinguished from other previous works in the sense of using automated red-teaming methods instead of manual red-teaming.
> > >
> > > However, there is already a line of research that has attempted to construct data automatically using a language model [3,4]. Furthermore,  I noticed that Lee et al., (2023) [4] have already proposed a toxic data construction method that employs an automated red teaming approach. Please refer to Figure 1 in their paper.
> > >
> > > This leads me to a point of confusion regarding your definition of "automated red teaming." Hence, I would appreciate an explanation as to why you categorize the method used in [4] as a manual red-teaming approach in your related work section. Also, can you clarify the differences between your work and [4]?
> > >
> > > - [1] Build it Break it Fix it for Dialogue Safety: Robustness from Adversarial Human Attack, Dinan et al., 2019.
> > > - [2] Bot-Adversarial Dialogue for Safe Conversational Agents, Xu et al., 2021.
> > > - [3] WANLI: Worker and AI Collaboration for Natural Language Inference Dataset Creation, Liu et al., EMNLP findings 2022.
> > > - [4] SQuARe: A Large-Scale Dataset of Sensitive Questions and Acceptable Responses Created Through Human-Machine Collaboration, Lee et al., ACL 2023.

---

> > > > ### Author Response · Authors · 2023-11-21
> > > > **Thanks + def of "automated red teaming" and differences with [3,4]**
> > > >
> > > > This reply is clarifying and helpful; thank you.
> > > >
> > > > The type of "automated red teaming" method that we focus on is the type of process that the first intro paragraph describes. It takes as input some operationalization of bad text (i.e. examples or a classifier) and outputs a set of adversarial examples. This is essentially the same thing as our Exploit step. The methods from the "Red-teaming with automated searches for natural language prompts" paragraph of the related works section all fall into this category. We are updating the introduction to make it clear that this is our focus.
> > > >
> > > > Regarding the differences between works like [3,4], the key one is that the human is outside the loop in our method (doing Explore/Establish) but inside the loop to annotate/label data in [3,4]. Another way of describing the difference is that the step that produces adversarial examples in our case (the Exploit step) does not involve a human, while their adversarial data generation does. Both types of methods are clearly valuable, but one useful property of our type of approach is that a fixed amount of human effort can then allow for highly scalable automated adversarial example generation. We are adding to the introduction section an explanation of this point as well.
> > > >
> > > > Thanks again for your time. We have found your feedback to be directly helpful.
> > > >
> > > > Best,
> > > >
> > > > -authors

---

> > > > > ### Author Response · Authors · 2023-11-22
> > > > > **Updates to intro**
> > > > >
> > > > > Feel free to look at the new first and third paragraphs there. We are glad you pointed out that we were clear about the relationship of our work to human-in-the-loop red teaming.
> > > > >
> > > > > Excerpts:
> > > > >
> > > > > > Automated attacks are valuable for red teaming, but they require that the harmful behavior can be identified efficiently beforehand. For instance, Perez et al. (2022b) depend on a pre-existing toxicity classifier, and Zou et al. (2023) use specific, user-provided phrases as target outputs. But this is often unrealistic. Usually, a red team must work from a more abstract specification and tailor their work to a specific application. For example, ethical norms are highly contextual (Schmidt & Wiegand, 2017; Dinan et al., 2019; Hendrycks et al., 2020; Xu et al., 2021)...
> > > > >
> > > > > > In this work, we introduce a red-teaming framework that uses automated attack tools but does not assume that the red team starts with an efficient way to identify failures. Instead, they must work from an abstract specification of undesired behavior. Figure 1 illustrates our approach. It requires a fixed amount of human effort outside the loop and leverages automated attacks to generate examples scalably. This allows for more flexibility than prior automated red-teaming methods while being more scalable than human-in-the-loop methods...

---

> ### Comment · Reviewer_tVNe · 2023-11-22
>
> Thank you for the update. The revised introduction appears to be clearer in conveying the contribution of your work. However, I still have difficulty understanding which point of your work is more beneficial than [1]. I have some additional questions for clarification.
>
> In your previous responses, you commented that:
> ```
> Both types of methods are clearly valuable, but one useful property of our approach is that a fixed amount of human effort can then allow for highly scalable automated adversarial example generation.
> ```
> Can you provide more details or evidence to support this statement?
>
> Based on your responses, the primary distinction between your work and [1] appears to be the presence of annotation during the exploitation step, with your approach offering a lower annotation cost compared to previous *partially automated human-in-the-loop* methods [1,2]. However, [1] addressed the problem of high annotation cost by only annotating candidates in the gray area, where filter predictions are ambiguous. Also, it is worth noting that [1] also can be *fully automated* by stopping annotation and using only filter predictions to label candidates.
>
> In this regard, if you were to allocate the same annotation budget in both approaches, can you clarify why your approach is beneficial than [1] in such a scenario?
>
> [1] SQuARe: A Large-Scale Dataset of Sensitive Questions and Acceptable Responses Created Through Human-Machine Collaboration, Lee et al., ACL 2023.
> [2] WANLI: Worker and AI Collaboration for Natural Language Inference Dataset Creation, Liu et al., EMNLP findings 2022.

---

> > ### Author Response · Authors · 2023-11-22
> > **An example**
> >
> > Thanks! Suppose that we want to produce $n$ adversarial prompts for a language generator. If we want to do it with a human **in** the loop, then regardless of whether the human is needed to produce/screen *all* examples or only a *fraction* of them, we will need to have $O(n)$ examples worth of human effort. This is like [1], [3], or [4]. On the other hand, if we have a human **out** of the loop with our technique, we will only need a constant amount, $O(1)$, of human involvement. Plus, our technique produces a new classifier and dataset as a byproduct.
> >
> > This illustrates one particular advantage of using a human out-of-the-loop in our case. In reality, though, different techniques produce qualitatively different attacks, and different methods will have different strengths and weaknesses in different circumstances. So the goal for red teaming research is working toward a versatile toolbox instead of some best overall method.
> >
> > Best,
> >
> > -authors

---

### Official Review · Reviewer_FuAh · 2023-10-29

**Soundness:** 3 good
**Presentation:** 4 excellent
**Contribution:** 3 good
**Rating:** 8
**Confidence:** 4

**Summary:**

This paper presents a new LLM red-teaming methodology with three steps: explore, establish, and exploit. The purpose of this method is to enable red-teaming of LLMs in cases where there is no pre-existing understanding of what kinds of outputs would be considered “bad” for the model. In the first step, the method samples prompts and outputs as an exploratory stage. Then, using humans in the loop, examples from the previous stage are labeled and a model and task specific classifier is trained with a label set defined based on the types of outputs seen in the explore stage. Finally, the exploit stage uses reinforcement learning and the output classifier to train an LLM capable of generating adversarial prompts for the LLM being red-teamed. In two experimental settings, the paper shows results that suggest the method does allow for improved generation of prompts that elicit forbidden outputs. The paper also introduces a dataset, CommonClaims, that contains statements that are labeled as common-knowledge-true, common-knowledge-false, or neither.

**Strengths:**

The primary strength of the paper is in the novel method it introduces. The work systematically breaks down a sensible approach to red-teaming an LLM, and the results seem to indicate that it works reasonably well. The paper is very clearly written and each step of the proposed method is motivated and explained well. Overall, the work is a very solid contribution and fills a gap in the LLM evaluation literature.

**Weaknesses:**

The primary weaknesses of the paper are its limited evaluation and the discussion of the approach’s limitations overall. The paper tests out the red-teaming approach on two different models, both GPT based, to attempt to elicit the model to produce toxic or false statements. Both of these are targeted at GPT-based models (GPT-2 and GPT-3). It would have been nice to see evaluations on other common LLMs, such as Bard, Llama, or Claude. This is only a mild weakness of the work, however, because the paper is mostly about introducing the new method. The second weakness is just that I would have liked to see more discussion of the limitations of the approach at a high-level. For example, are there alternatives to using an LLM to generate adversarial prompts? What kinds of biases might this introduce? How scalable is the method when the second step requires human input? The lack of discussion of these questions is not a major issue, but it would be nice in a future version to see the limitations section fleshed out a bit more.

**Questions:**

The paper mentions that human input can be used to determine the set of labels used to train the classifier in step 2 of the method. Can the authors describe a bit more what that process looks like in practice?

Table 4 shows some examples of completions. I notice that some of them seem a bit nonsensical or have strange spacing and grammar in places (for example, the completion that ends in ButTONIC). Can the authors expand on why this happens, and how does this affect the evaluation of which outputs are problematic?

---

> ### Author Response · Authors · 2023-11-16
> **Response to FuAh**
>
> Thank you for the feedback and comments. We are glad to hear that you found the paper to be well-written and to address an important gap in the literature. Below are replies to weaknesses in order.
>
> ### W1 – Re: Concerns about only using GPT-2 and GPT-3
>
> **We agree** that working with Llama, Claude, and Bard-type models would be useful for establishing clearer relevance. We added this to our discussion of future work.
>
> **We also emphasize that our method is black-box, and the RL attacks that we use in the exploit step could be replaced with other attack methods.** In the adversarial attacks literature, findings with individual methods/models often quickly become outdated, but these two facts make us expect that our work may have some implications that may continue to be valuable further in the future.
>
> ### W2 – Re: adding to the discussion of limitations
>
> **We have added to Section 2, and Section 5 to discuss these.** Thank you for pointing these out. We discuss how alternatives could have been used other than the RL attack we used (see the “Red-teaming with automated searches for natural language prompts” paragraph). We added references to works on AI oversight ([Bai et al., 2022](https://arxiv.org/abs/2212.08073)), AI feedback ([Lee et al., 2023](https://arxiv.org/abs/2309.00267)), active learning ([Zhen et al., 2022](https://arxiv.org/abs/2203.13450)), and weak supervision ([Boecking, 2020](https://arxiv.org/abs/2012.06046)) to discuss how human involvement might be effectively scaled.
>
> ### Q1 – Re:  “human input can be used to determine the set of labels used to train the classifier…Can the authors describe a bit more what that process looks like in practice?”
>
> **Abstractly, the establish step can be divided into three substeps.** After the red team has expired the data, they pick a label set, obtain labels, and then train a classifier for harmful text on those labels. So in our case, the answers that we obtained from crowdworkers were not used to select the label set. The label set was selected by us after our own analysis of the data.
>
> **Consider how this compares to prior approaches.** Naively (e.g. without developing a contextual understanding of failure modes), it may seem reasonable to use a classifier trained on the CREAK dataset of true and false statements. But as we find, this does not work well, presumably because of the distributional differences and the lack of a “neither” label.
>
> ### Q2 – Re: Concerns about dysfluencies in prompts
>
> **We agree that the results in Table 4 have dysfluencies and repetition, but we emphasize two points, and we updated the discussion of these results accordingly.**
> - Prompts appearing as fluent English is not required for them to be adversarial. For example, [Zou et al., 2023](https://arxiv.org/abs/2307.15043) studied jailbreaks with triggers such as “restored into one sentence grammar using proper colon”. We only focus on developing diverse and adversarial prompts. Making plain-English ones is not a goal of ours, however, it would be possible to do with a stronger KL penalty on the generator’s outputs.
> - We agree that there is room to further improve the diversity of the attack distribution. However, our method is much better than the baseline. Qualitatively, the results in Table 4 are much more diverse than the results from the baseline in Table 6. And without our diversity reward, 61 out of the 100 sentences we sampled were all identical.
>
> Thank you again for your time and help! We are looking forward to the rest of the discussion period.

---

### Official Review · Reviewer_weY9 · 2023-11-03

**Soundness:** 3 good
**Presentation:** 3 good
**Contribution:** 2 fair
**Rating:** 3
**Confidence:** 4

**Summary:**

The paper describe a framework for redteaming a LLM from scratch, that is, it consists of finding possible behaviour problems of the model, labelling the vulnerabilities and finding malicious or adversarial prompts that will elicit such undesirable behaviour.

The proposed method is just a compilation of relevant known techniques from the literature. The set of experiments are quite comprehensive (but still lack comparison) and the results (although shown against GPT3-davinci-002) indicates the importance of dealing with this problem. Overall, this paper describes a good engineering solution to an important problem.

**Strengths:**

1. The paper proposes an effective solution to a very important problem to find prompts that will generate undesirable contents.

2. The proposed methods are all plausible and easy to apply in a similar settings in practice.

3. The evaluation is quite good but lack comparative studie and many sane and helpful conclusions are drawn.

**Weaknesses:**

1. The proposed method is claimed to be mainly different from the previous work in that it has two more steps: exploration and establishment. However, these two steps are straight-forward (e.g., using existing diversification technique to explore the output space) or still mainly rely on human annotation (interaction) (e.g., the "establish" step). Therefore, it is not essentially different or more challenging than what was for the previous work.

2. For the problem setting considered by the paper (i.e., from scratch), the exploration step may be the most critical. The current proposal require internal state information of the LLM, and hence cannot be used in close-source or API-only LLMs, which limits the potential impact of this study.

3. It is desirable to compare with previous work and other adversarial prompt generation techniques (in a as fair setting as possible) to better evaluate the performance of the proposed method. Currently, this is unknown.

4. The generated adversarial prompts seem to have some repeating words from time to time and also seem to be not that diversified. Is it the nature of the problem or some artifact of the method?

**Questions:**

See Weaknesses

---

> ### Author Response · Authors · 2023-11-16
> **Response to weY9**
>
> Thank you for the feedback and comments. We are glad to hear that you found the paper to contribute an effective solution to a very important problem. We do our best to address your concerns below. Please let us know if there is anything else that we can do to help.
>
> ### W1 — Re: Straightforwardness of the method
>
> **We agree that this is a natural way to approach the problem. However, there has not been prior work to show the benefits of a contextually defined target for automated red teaming.** Prior works on red teaming have tended to neglect the role of preference formation and human factors in AI oversight, and simple filtering-based baselines. To the best of our knowledge, ours is the first work on automated attack methods that addresses these.
>
> **The Explore and Establish steps were the reason why our red teaming was more successful than the experiment with the pre-existing CREAK data.** Based on prior work (e.g. [Perez et al. (2022)](https://arxiv.org/abs/2202.03286)) one might suspect that the best way to approach automatic red-teaming for false statements is to use an off-the-shelf classifier for true vs false statements. However, we show that using a classifier trained on the CREAK dataset, was not effective (see Section 3.2.1 and Appendix D). Contextual red teaming was needed.
>
> ### W2 – Re: Concerns about using the target model’s latents
>
> **Sorry for the lack of clarity – we updated Section 2 and Figure 2 to clarify that we do not need the target model’s latents.** We apologize that we were unclear about whether we need the target model’s latents. We only use these when available. We have updated our description of the methods to say that when latents are not available, we use another model’s embeddings as done in Section 3.2. Internal activations of GPT-3-text-davinci-002 were not available via API, so we instead used embeddings from GPT-3-text-ada-002, a dedicated text encoder.
>
> ### W3 – Re: Making comparisons to other methods
>
> **We want to clarify that we compare our method to conventional RL-based attacks and quantitatively analyze the results.** We improve on the RL attack method from [Perez et al. (2022)](https://arxiv.org/abs/2202.03286) and [Deng et al. (2022)](https://arxiv.org/abs/2205.12548) with our approach to producing diverse adversarial prompts. In our control experiments that did not include the diversity technique, the prompt generators exhibited mode collapse. We are sorry that this was not clear, and we updated our description of these experiments to point out that this was the method used by these two prior works. Is there another experiment that you would recommend that we run?
>
> **Comprehensive benchmarking is important, but out of scope.** In this paper, our goal was to run experiments to validate the benefits/performance of the improvements that we make to red teaming and RL attacks. We agree that other approaches to automated red teaming are interesting to compare, but we think that this would best be explored in separate work. We note that there currently are [two](https://trojandetection.ai/) [competitions](https://satml.org/participate-competitions/) being run to do this, and we are contributing to this by testing our RL attacks with one of them.
>
> **We have added a discussion of benchmarking to our future work section and clarified the goal of our experiments/evaluations.**
>
> ### W4 – Re: Concerns about dysfluencies in prompts
>
> **We agree that the results in Table 4 have dysfluencies and repetition, but we emphasize two points, and we updated the discussion of these results accordingly.**
> - Prompts appearing as fluent English is not required for them to be adversarial. For example, [Zou et al., 2023](https://arxiv.org/abs/2307.15043) studied jailbreaks with triggers such as “restored into one sentence grammar using proper colon”. We only focus on developing diverse and adversarial prompts. Making plain-English ones is not a goal of ours, however, it would be possible to do with a stronger KL penalty on the generator’s outputs.
> - We agree that there is room to further improve the diversity of the attack distribution. However, our method is much better than the baseline. Qualitatively, the results in Table 4 are much more diverse than the results from the baseline in Table 6. And without our diversity reward, 61 out of the 100 sentences we sampled were all identical.
>
> Thank you again for your time and help! We are looking forward to the rest of the discussion period.

---

### Official Review · Reviewer_8GJm · 2023-11-03

**Soundness:** 3 good
**Presentation:** 3 good
**Contribution:** 3 good
**Rating:** 5
**Confidence:** 4

**Summary:**

This paper proposes a three-step framework  for red-teaming “from scratch” in which the adversary does not begin with a way to classify failures. They also construct the CommonClaim dataset of 20,000 statements labeled by humans as common-knowledge-true, common knowledge-false, or neither.

**Strengths:**

1. Implementation of each step are introduced in details in Section 2 (Method) and 3 (Experiment)
1. The design of three steps in the proposed framework is clearly described in Section 2.

**Weaknesses:**

1. Red teaming normally uses manual or automated methods to adversarially probe a language model for harmful outputs, and then updates the model to avoid such outputs. However, in this work, only some case studies have been conducted to show the proposed framework is able to generate prompts elicit harmful content from LLMs. The work would be more complete if more quantitative results are presented and the follow-up model update is accomplished.

2. Some other red teaming methods are not compared, e.g.,

Perez, Ethan, et al. "Red teaming language models with language models." arXiv preprint arXiv:2202.03286 (2022).

3. There is no ablation study to justify the effectiveness of the design of each step in the proposed framework

**Questions:**

1. Although it is said that the data and code are available, not the actual location to fetch those resources is not provided. In the supplementary materials, only the code is included.

2. According to the abstract and introduction section, the goal of this work is to red-team from scratch. However, in step 2 of the proposed framework, we still need to choose a label set such that one of the labels represents undesirable outputs.  This indicates the category of undesirable output is pre-defined, which is inconsistent with the goal of this work.

---

> ### Author Response · Authors · 2023-11-16
> **Response to 8GJm**
>
> Thank you for the feedback and comments. We are glad to hear that you found the design and implementation of our experiments to be good. Here are replies to weaknesses and questions in order.
>
> ### W1 – Re: Adversarially training the target model after attacking it.
>
> **We agree that adversarial training is an important next step.** In the “What comes after Explore/Establish/Exploit?” paragraph, we discuss this and others. Enabling this is a motivation of ours.
>
> **Our contributions are focused on red teaming but we believe they are sufficient and valuable in their own right.** Please see our global response for a summary of contributions, quantitative results, and qualitative results. Prior works on red teaming such as [Perez et al. (2022)](https://arxiv.org/abs/2202.03286) have not performed adversarial training either. Meanwhile, our work is the first to our knowledge to use automated attacks to red team a model  of a size comparable to GPT-3 for untruthful text. We also release the CommonClaim dataset. We updated the “Future work” paragraph to discuss how we do not perform any outer applications of the dataset, classifier, or adversarial data.
>
> ### W2 – Re: Comparisons to other methods such as Perez et al. (2022a).
>
> **We want to clarify that we compare our method to the RL-based one used by [Perez et al. (2022)](https://arxiv.org/abs/2202.03286) and quantitatively analyze the results.** We improve on the RL attack method from [Perez et al. (2022)](https://arxiv.org/abs/2202.03286) and [Deng et al. (2022)](https://arxiv.org/abs/2205.12548) with our approach to producing diverse adversarial prompts. In our control experiments that did not include the diversity technique, the prompt generators exhibited mode collapse. We are sorry that this was not clear, and we updated our description of these experiments to point out that this was the method used by these two prior works. We also added vocabulary size analysis to better quantify mode collapse. Across 100 samples, the prompt vocabulary size decreased from 116 to 3 and 81 to 9 for the toxicity and untruthfulness red teaming experiments without our diversity method.  Is there another experiment that you would recommend that we run?
>
> ### W3 – Re: Ablation studies.
>
> **We want to clarify that we performed ablation experiments. We also have a new one underway.** Sorry that this was unclear. We updated our descriptions of these tests in the paper to clarify this. Thank you for this suggestion. The ablation tests were ones in which we removed the diversity term in the exploit step (Section 3.1.1, Section 3.2.1, Appendix B), replaced our false statement classifier with an off-the-shelf one (Section 3.2.1, Appendix D), and replaced our false statement classifier with one trained on chatbot labels (Section 3.2.1, Appendix E). We are currently running a new ablation test in which we omit the diversity subsampling from the Explore step. Please see the global response for all details. Are there other ablation tests you recommend we do?
>
> ### Q1 – Re: CommonClaim in supplement.
>
> **We updated the supplementary materials to include CommonClaim.** We apologize for not including this in the supplement. Thank you for pointing this out.
>
> ### Q2 – Re: “...the goal of this work is to red-team from scratch. However, in step 2 of the proposed framework, we still need to choose a label set…”
>
> **Could you please clarify this concern?** In our framework, the process of choosing a label set is how we define the target for the red team. For example, in our truthfulness red-teaming, we defined the categories as part of the process, based on interaction with the model outputs.
>
> **Our results show the benefits of contextual red teaming in comparison to a baseline that uses an off-the-shelf truthfulness classifier based on the pre-existing CREAK dataset.** When we red team for truthfulness in Section 3.2, we identified the need for a third additional category of “neither” true nor false to account for statements that were opinions, vague, obscure, uncommon knowledge, or otherwise hard to categorize as true or false by common knowledge. This third label is an advantage that our approach had over the control experiment with the CREAK dataset which used a classifier trained on a pre-existing dataset that did not include a “neither” category.
>
> Thank you again for your time and help! We are looking forward to the rest of the discussion period.

---

### Author Response · Authors · 2023-11-16
**Global Response**

We are thankful to all reviewers for their help and feedback. We are glad to hear that it was well-written & clear (8GJm, FuAh, tVNE), experiments were well-designed (8GJm, weY9, FuAh), that it contributes to solving a very important problem with red teaming research (weY9, FuAh), and that the CommonClaim dataset is valuable (tVNE).

In addition to individual responses, we would like to emphasize the contributions of the work. We also overview updates to the paper and respond to common points raised.

### Contributions and key results

1. **We identify a gap with existing research on automated red teaming methods and propose a new framework (EEE: Explore, Establish, Exploit) for more realistic red teaming.** A number of prior works use have used a classifier for harmful text without addressing where it comes from or taking into account simple filtering baselines. Our framework involves contextually refining the target behavior and can apply when a classifier for harmful outputs does not exist.
2. **We contribute a new objective for RL red teaming that reduces issues with mode collapse.** We demonstrate the value of this contribution in an experiment with a pre-trained toxicity classifier as the red-teaming target. With our method, toxicity increases by a factor of 30x compared to the baseline. Without our method, RL undergoes mode collapse and fails to elicit toxic text.
3. **We apply the EEE framework to red team GPT-3 to produce false statements.** To the best of our knowledge, we are the first to successfully perform red teaming for untrue text against a language model at this scale using an automated method.
4. **Among other control experiments, we show that prior approaches (e.g. [Perez et al. (2022)](https://arxiv.org/abs/2202.03286), [Deng et al. (2022)](https://arxiv.org/abs/2205.12548)) do not succeed at red teaming GPT-3 for false statements.**
    - Without our diversity method, the adversarial prompt generator undergoes mode collapse.
    - Without contextual red teaming, a pretrained off-the-shelf classifier provides a hackable reward signal.
5. **We contribute the CommonClaim dataset** of 20k factual claims labeled as common-knowledge true, common-knowledge false, or neither by humans.

### Quantitative results

- Increasing toxicity by over 30x in toxicity experiment according to the classifier we used in this experiment to stand in for a human (Section 3.1.1).
- Avoiding mode collapse with our intra-batch diversity technique
    - GPT-2-xl: Across 100 sampled prompts, the prompt generator trained without the diversity term used a total of 3 distinct vocabulary words (compared to 116 for the diverse model) and had a 0% toxicity rate. (Section 3.1.1 and Appendix B).
    - GPT-3-davinci-002: Across 100 sampled prompts, the prompt generator trained without the diversity term used a total of 9 distinct vocabulary words (compared to 81 for the diverse model) and generated the exact same sentence in 61 of 100 samples. (Section 3.2.1 and Appendix B).
- Analysis of human and ChatGPT label agreement on CommonClaim (Table 3) and intra-labeler consistency (Section 3.2).
- Analysis of how our classifier, the CREAK classifier, and the ChatGPT-label classifier labeled CommonClaim (Section 3.2.1, Appendix D, and Appendix E).
- Untrue statements increased from 30% to 74% according to our classifier in experiments with GPT-3-text-davinci-002 (Section 3.2.1).
- Analysis of how in the CREAK and ChatGPT control experiments the prompt generators succeeded in generating “untrue” text according to the classifiers for those experiments (Section 3.2.1, Appendix D, Appendix E).

### Qualitative results

- The prompt generators for the toxicity experiments learned to discuss sensitive topics (Table 1 and Section 3.1.1).
- The prompt generators for the truthfulness experiments learned to discuss politics (Table 4 and Section 3.2.1).
- The prompt generators trained with our diversity method exhibited less mode collapse than the ones trained without. Across 100 samples, the prompt vocabulary size decreased from 116 to 3 and 81 to 9 for the toxicity and untruthfulness red teaming experiments without our diversity method. (Section 3.1.1, Section 3.2.1, Appendix B, Table 1 versus Table 5, and Table 4 versus Table 6).
- The prompt generators for the CREAK dataset control experiment showed no apparent tendency to elicit anything related to untrue statements (Table 7, Appendix D).
- The prompt generators for the ChatGPT label dataset control experiment showed no apparent tendency to elicit anything related to untrue statements (Table 8, Appendix E).

---

> ### Author Response · Authors · 2023-11-16
> **Global Response Part II**
>
> ### Ablation experiments/baselines
>
> - To measure the impact of our diversity technique, we ablated the diversity term in the toxicity and truthfulness experiments. We observed mode collapse without this term. See above  (Section 3.1.1, Section 3.2.1, Table 5, Appendix B).
> - To compare our contextual EEE approach to one that simply uses a classifier off the shelf, we compare using our untrue text classifier to one trained on the pre-existing CREAK dataset. We find that the CREAK classifier fails to elicit anything related to untrue claims (Section 3.2.1, Appendix D, Table 4, Table 7).
> - To compare human labels to chatbot labels, we compare using our untrue text classifier trained on human labels to one trained on GPT-3.5 labels. We find that the classifier trained on chatbot labels fails to elicit anything related to untrue claims (Section 3.2.1, Appendix E, Table 4, Table 8).
>
> ### We are running a new experiment – results to come soon
>
> To test the influence of diversity subsampling in the Explore step, we are rerunning the full toxicity red teaming experiment (all 3 steps) without diversity subsampling in the first step. Thank you to tVNe for pointing this out. Updates to follow.
>
> ### Updates to the paper and supplemental materials (details in individual responses)
>
> - Updating the description of our contributions in the introduction.
> - Updates to the limitations and future work paragraphs in response to all four reviewers.
> - We updated Section 2 and Figure 2 to clarify that our method does not rely on white-box access to the target model. When its latents are not available to produce embeddings (i.e. the experiments in Section 3.2), we use another embedding model instead. Thank you to weY9 for pointing this out.
> - We added vocabulary size analysis to better quantify mode collapse. Across 100 samples, the prompt vocabulary size decreased from 116 to 3 and 81 to 9 for the toxicity and untruthfulness red teaming experiments without our diversity method.
> - Adding CommonClaim to the supplemental materials in common_claim.csv. Thank you to weY9 for pointing this out.
> - Expanding on our discussion of dysfluencies in the generated prompts. Thank you to FuAh for pointing this out.
> - Adding discussion of several related works. Thank you to tVNe for pointing this out.

---

### Meta-Review · Area_Chair_zVj2 · 2023-12-07

**Metareview:**

The authors introduce a framework for red-teaming an LLM "from scratch", i.e., in scenarios in which it is not assumed to be known a priori what kinds of harmful or undesirable content might be generated by such models. There was consensus amongst reviewers that addressing assumptions implicit in red-teaming LLMs is an interesting direction, and the work offers an intuitive framework toward this end.

The main novelty on offer here are the first steps in the framework (explore, establish); once this is done, existing strategies can be adopted for "exploitation". One issue here with the contribution, then, is that these steps are both intuitive and rather open-ended, and therefore it is difficult to assess how much of a contribution making these explicit is (a point made by weY9), especially in light of existing related work on similar "human-in-the-loop" frameworks (tVNe). The back-and-forth during the response period concerning this latter point, especially, suggests the authors could do a better job of distinguishing this "framework" from existing related work.

**Justification For Why Not Higher Score:**

Primarily, the contribution here vis-à-vis existing work is not entirely clear, and the "framework" on offer is so high-level that it is difficult to gauge its potential impact; so much would depend on its instantiation.

**Justification For Why Not Lower Score:**

N/A

---

### Decision · Program_Chairs · 2024-01-16

Reject